# Involvement of CXCR4 in Normal and Abnormal Development

**DOI:** 10.3390/cells8020185

**Published:** 2019-02-20

**Authors:** Nanako Kawaguchi, Ting-Ting Zhang, Toshio Nakanishi

**Affiliations:** Department of Pediatric Cardiology, Tokyo Women’s Medical University, Tokyo 162-8666, Japan; 13572273657@163.com (T.-T.Z.); nakanishi.toshio@twmu.ac.jp (T.N.)

**Keywords:** CXCR4, CXCL12, SDF-1, cancer, pulmonary hypertension, stem cell

## Abstract

CXC motif chemokine receptor type 4 (CXCR4) is associated with normal and abnormal development, including oncogenesis. The ligand of CXCR4 is stromal cell-derived factor (SDF), also known as CXC motif ligand (CXCL) 12. Through the SDF-1/CXCR4 axis, both homing and migration of hematopoietic (stem) cells are regulated through niches in the bone marrow. Outside of the bone marrow, however, SDF-1 can recruit CXCR4-positive cells from the bone marrow. SDF/CXCR4 has been implicated in the maintenance and/or differentiation of stemness, and tissue-derived stem cells can be associated with SDF-1 and CXCR4 activity. CXCR4 plays a role in multiple pathways involved in carcinogenesis and other pathologies. Here, we summarize reports detailing the functions of CXCR4. We address the molecular signature of CXCR4 and how this molecule and cells expressing it are involved in either normal (maintaining stemness or inducing differentiation) or abnormal (developing cancer and other pathologies) events. As a constituent of stem cells, the SDF-1/CXCR4 axis influences downstream signal transduction and the cell microenvironment.

## 1. Introduction

Chemokines are a large family of small, structurally related cytokines. There are two major subgroups of chemokines characterized by the arrangement of their first two cysteines, either adjacent to one another (CC subfamily) or separated by one amino acid (CXC subfamily). So far, 47 chemokines and 20 chemokine receptors have been identified in humans [1]. CXC motif chemokine receptor type 4 (CXCR4) is a G protein-coupled receptor that is involved in homing and chemotaxis in the hematopoietic and immune systems. The ligand of CXCR4 is a stromal cell-derived factor (SDF)-1/CXCL12 (hereafter referred to as SDF-1) that is associated with stem cell migration from the bone marrow to the cells expressing SDF-1 [2]. SDF-1 is highly expressed in bone marrow stromal cells but it is also widely expressed in diverse organs including the brain, heart, liver, and lungs. It has been suggested that SDF-1 and CXCR4 are involved in the recruitment of stem cells that differentiate into the cells necessary to repair tissue damage [3,4]. One example of this is the finding that gene therapy using SDF-1-overexpressing cells (skeletal myoblasts and cardiac fibroblasts) can increase c-kit-positive stem cells in the area of an infarct [5]. Additionally, direct injection of an expression vector containing SDF-1 reportedly improved cardiac function and increased vessel density in an infarcted heart [6,7]. SDF-1-over-expressing bone marrow stem cells could form cardiac stem-like depolarizing cells in the infarct area in vivo that function as immature cardiomyocytes [8]. Interestingly, SDF-1 is one of the significant genes selected from the genome-wide association studies of coronary artery disease, although the mechanism of action remains unclear [9,10]. The repair function of the SDF-1/CXCR4 axis may help to avoid disease.

Of note, Imitola et al. discovered that the migration of neural stem cells (NSCs) is mediated by the SDF-1/CXCR4 pathway during injury and inflammation. They noted that SDF-1/CXCR4 contributes to migration not only in bone marrow cells as previously described, but also in other cell types such as NSC [3]. This idea was novel in that inflammation was associated with recruiting stem cells that would differentiate into the required cells. The observed amelioration of the infarct area as a result of the injection of SDF-1 supports this idea. 

CXCR4 has been associated with several diseases, including human immunodeficiency virus (HIV) infection, cancer, warts, hypogammaglobulinemia, immunodeficiency, myelokathexis (WHIM) syndrome, pulmonary artery hypertension (PAH), and pulmonary injury. Several CXCR4 inhibitors have been developed to treat these diseases, especially cancer. Plerixafor is a CXCR4 antagonist that was approved by the Food and Drug Administration (FDA) in 2008. Many other CXCR4 antagonists that bind the CXCR4 receptor transmembrane domain have been developed [11]. In this review, we present evidence for the involvement of CXCR4 in these complicated conditions to either maintain normal function or promote malfunction.

## 2. CXCR4 Involvement in Normal Development

### 2.1. CXCR4 Knock-Out Studies

CXCR4 was originally identified as the leucocyte-expressed seven-transmembrane domain receptor (LESTR)/fusin, a co-receptor for HIV. Recent reports concerning gene editing of CXCR4 demonstrated inhibition of HIV infection [12,13,14], which may prove to be a fundamental treatment for acquired immunodeficiency syndrome. The ligand for this receptor is SDF-1 [15,16,17,18]. SDF-1 exists as the isoforms 1α, 1β, and 1γ. A strong sequence conservation exists between mouse and human SDF-1 (99%), but not in monocyte chemotactic protein (MCP)-1 (55%) nor in macrophage inflammatory protein (MIP)-1α (75%) [18].

SDF-1 can activate CXCR4-expressing cells and induce B cell proliferation and T cell recruitment. The phenotypes of the knock-out mice for the receptor and the ligand, CXCR4^−/−^ and SDF-1^−/−^, respectively, are similar, and the defect is observed in hematopoiesis. Notably, few mature B cells and T cells have been observed within the peripheral lymphoid organs [19,20]. In addition, CXCR4^−/−^ mice display impaired vascularization in various organs, including the intestines, stomach, and heart. The latter is a ventricular septal defect that occurs during embryogenesis [21]. Additionally, a defect was observed in cerebellar development [19]. SDF-1 is highly conserved (64–66% identical among mammalian sequences) and is similarly expressed among mammals [22]. Studies using knock-out mice have revealed the involvement of CXCR4 in cardiogenesis and brain development. Without CXCR4, the homozygous knock-out mice die before birth. Therefore, CXCR4 is a key molecule for normal development.

DiGeorge, or 22q11.2-deletion syndrome (DGS), is the most common human genetic deletion syndrome, and this disease results in mental disorders, craniofacial dismorphogenesis, thymus hypoplasia, and congenital heart defects caused by cardiac outflow anomalies. The responsible gene is located within 22q, which also includes T-box-containing transcription factor (TBX) 1. *TBX-1* is one of the major genes responsible for forming the pharyngeal arch-derived structure [23,24,25,26]. Neural crest (NC) cells migrate from the dorsal hindbrain toward arches and ultimately form the face, neck, and chest. Additionally, NC cells contribute to cardiovascular structure. Therefore, DGS was believed to result from defective NC development; however, the *TBX-1* gene is not expressed in NC cells in pharyngeal arches. *TBX-1* is expressed in pharyngeal ectoderm, endoderm, and mesoderm. The interactions of pharyngeal NC cells and the surrounding tissues such as pharyngeal ectoderm, endoderm, and mesoderm were studied in the context of TBX-1 signaling. Fibroblast growth factor 8, vascular endothelial growth factor, and more recently, SDF-1, were shown to be involved in this interaction [27,28]. Notably, the phenotype of an SDF-1/CXCR4 knock-out mouse is similar to that of a mouse exhibiting DGS, with defects in heart development and mental retardation [29]. Interestingly, SDF-1-expressing cells are present in the outflow track, while CXCR4^+^ cells are present in the NC. Thus, CXCR4 may be involved in the migration of NC cells during normal development. Recently, however, SDF-1/CXCR4 was suggested to function in cardiovascular development in the context of second heart field to endothelial cells and not in TBX-1 haplosufficient arch artery phenotype [30]. More studies are required to clarify CXCR4 contribution to NC cells in the context of normal development. 

### 2.2. SDF-1/CXCR4 Functions in Tissue Repair

SDF-1 is strongly expressed in bone marrow stromal cells. SDF-1 recruits stem cells and regulates their differentiation to repair injuries. Repair is mediated by growth factors and cytokines that include transforming growth factor-beta (TGF-β) [31] and vascular endothelial growth factor (VEGF) [32] in the damaged tissues. SDF-1 is a chemokine peptide and can be inactivated quickly by matrix metalloproteinase-2 (MMP-2), which is abundant under inflammatory conditions. Wang et al. used a gene-activated collagen (GAC) substrate to sustain cellular expression and found that local cellular expression of the SDF-1 α gene induces isolated c-kit^+^ stem cell homing to collagen matrices [33]. Additionally, they transplanted a GAC-coated membrane into the quadriceps of mice and found that only SDF-1 GAC, and not GFP GAC, could recruit c-kit-positive cells, suggesting that SDF-1 specifically promotes c-kit^+^ stem cell homing. Mesenchymal stem cells (MSCs) promote proliferation and differentiation of c-kit^+^ cardiac stem cells via SDF-1/CXCR4 signaling. If treated with the CXCR4 antagonist AMD3100, cardiac stem cells derived from a murine postnatal cardiac explant differentiate into cardiac myocytes [34]. Therefore, MSCs may control the self-renewal and/or proliferation of c-kit^+^ cardiac stem cells through SDF-1/CXCR4 signaling [34]. The c-kit^+^ cardiac stem cells are thought to rescue cardiac damage from injury such as ischemia [35,36,37]. Contradictory data, however, draw into question if the c-kit^+^ cells can act as cardiac stem cells [38,39]. When c-kit^+^ cells were utilized for cardiac repair, there were positive effects on cardiac differentiation and the enhancement of cardiac myocyte survival [40,41,42,43]. SDF-1/CXCR4 can act as an upstream regulator of these c-kit^+^ cells and may contribute to the repair of tissues directly [44] or indirectly through the action of MSCs, as previously described [34]. 

Similarly, neural stem/progenitor cells are controlled by SDF-1/CXCR4 to maintain stemness [45]; however, CXCR4 activation promotes the differentiation of human embryonic stem cells into NSCs [46], suggesting that CXCR4 may work to stabilize NSCs. This form of up- or down-regulation to maintain certain populations such as NSCs may also occur in other stem cell populations. 

It is established that after pneumonectomy, the left lung possesses greater capacity for respiratory function due to alveolar regeneration. Recently, platelets were reported to be involved in this regeneration through SDF-1. Without platelets, which are induced by thronbopoietin-deficient conditions (5% of wild type), regrowth of the lungs reduced markedly. SDF-1 could enhance this recovery significantly. Additionally, SDF-1^+/+^ platelets could rescue lung regeneration, while SDF-1^−/−^ platelets could not. Thus, pneumonectomized mice need SDF-1 from platelets for alveolar regeneration [47]. The recent idea that the lung features hematopoietic niches for hematopoietic stem cells is noteworthy [48,49]. Further studies focused on the involvement of SDF-1 and CXCR4 in regeneration could provide insight into their roles in this process. The liver is also thought to possess niches formed by hepatic stellate cells [50]. The niche is on the basement membrane surrounded by the matrix (laminin and collagen type IV). The stellate cells can act as myofibroblasts that form a niche in the bone marrow and can synthesize matrix. Interestingly, Jagged-1 which is often expressed in the niche, is also expressed in these hepatic stellate cells. Of note, the stellate cells can be stem cells [50]. The results suggest a complicated flexible signature of the stem cell-containing niches.

## 3. CXCR4 Involvement in Abnormal Pathological Development

### 3.1. CXCR4 in Cancer

In contrast to its normal role in stem cell function during development, CXCR4 is also expressed in various cancers [51]. A recent, relatively large-scale polymorphism analysis (3684 cancer patients and 5114 healthy controls) suggested an association between high CXCR4 expression and increased risk of cancer [52]. Another study presented correlated high CXCR4 expression with decreased survival in patients with breast carcinoma [53]. Additionally, cells expressing CXCR4-Δcarboxy terminal domain (CTD) can exhibit epithelial–mesenchymal transition (EMT)-like changes, such as changes in stellate cell appearance, enhanced motility, and decreased cell adhesion, which are characteristic of tumor cells [54]. CXCR4 signaling increases stem cell longevity within the niches; however, this can increase the chances of DNA damage and a mutation being introduced into the stem cells. These cells may become cancer stem cells (CSCs) and ultimately promote metastasis and cancer progression. SDF-1 binding to the CXCR4 and CXCR7 receptors can activate divergent signaling through multiple pathways, such as the extracellular signal-regulated kinase (ERK) 1/2, stress-activated protein kinase/c-Jun N-terminal kinase (SAPK/JNK), AKT, mammalian target of rapamycin (mTOR), and Bruton tyrosine kinase pathways. This signaling can affect chemotaxis, proliferation, cell survival, and migration. Thus, alterations in signal transduction in normal stem cells can lead to the induction of a cancerous state. 

Various anticancer drugs that target the SDF-1/CXCR4 axis are under evaluation in clinical trials. Plexifor (AMD3100) induces mobilization of CD34+ cells in patients with multiple myeloma and non-Hodgkin’s lymphoma [55]. Its use, in combination with granulocyte colony-stimulating factor (G-CSF), could prove effective [56]. AMD3100 was approved by the FDA in combination with G-CSF to mobilize hematopoietic stem cells to the peripheral blood of patients with non-Hodgkin’s lymphoma [57]. Small molecules that antagonize CXCR4 have been developed [11], and some are in clinical trials. Currently, AMD3100 is most commonly used in combination with G-CSF to inhibit hematopoietic stem cell attachment to the bone marrow. A number of other CXCR4 antagonists have been developed [11]. A small cyclic peptide, LY2510924 [58], and an antibody against CXCR4, LY2624587 [59], have been developed to target tumors and cancers. An ongoing clinical trial is evaluating the effectiveness of LY2510924 for cancer treatment [60]. T134, a small-molecule CXCR4 inhibitor, binds to CXCR4 without cross-binding to AMD3100 [61]. Most of these small CXCR4 antagonistic molecules have been targeted to cancers. 

MSCs can contribute to cancer progression by becoming carcinoma-associated fibroblasts (CAFs) and are recruited to tumors through TGF-β and SDF-1 signaling [62,63]. CAFs express SDF-1 and Gremlin. Gremlin is not expressed in normal tissues. CAFs express alpha-smooth muscle actin (α-SMA), which is also expressed in myofibroblasts. Normally, myofibroblasts are niche cells present in the bone marrow, and these cells increase markedly in number during cancer progression to create niches for cancer cells that sustain cancer progression [63]. Quante et al. tracked α- SMA-red fluorescent protein (RFP) cells in transgenic mice that express RFP under the direction of the α-SMA promoter [63]. α-SMA-RFP-positive myofibroblasts could become CAFs after tumor infiltration. Therefore, myofibroblasts and CAFs may have the same origin, as both may be generated from MSCs [64]. The same authors described that following transplantation of human-leukemic cells, more cells stayed in the niche in the liver than migrated to the bone marrow or peripheral blood [64], suggesting that certain niches may be required for certain cancer/stem cells. The tumor microenvironment, particularly for CSCs, is similar to the niche for stem cells. Niche-constituted cells produce SDF-1 to protect and maintain CSCs. CSCs highly express CXCR4 and migrate to the highly expressed SDF-1-containing niche, and this migration may induce metastasis [65]. Thus, the SDF-1/CXCR4 axis is involved in metastasis through the microenvironment produced by the niche.

### 3.2. Pulmonary Arterial Hypertension (PAH)

Pulmonary arterial hypertension (PAH) is a rare disease characterized by high blood pressure in the arteries of the lungs, caused by an endothelial cell disorder and the abnormal proliferation of smooth muscle cells. These associate with inflammation and finally lead to occlusion within the pulmonary artery, which in turn triggers right ventricle hypertrophy. CXCR4^+^ and/or c-kit^+^ cells have recently been implicated in the development of PAH [66,67,68,69,70,71,72]. c-kit^+^ cells were observed in a rat model of PAH and are thought to be expressed by the smooth muscle lineage [66]. CXCR4 inhibitors have been reported to attenuate the development of PAH [70,71]; however, bone marrow stem cells overexpressing CXCR4 can function to repair the damage [73]. The contradictory nature of these results suggests that CXCR4 and stem cells can be associated with damage repair in addition to causing malignancy.

The CXCR7 inhibitor CCX771 reportedly attenuates PAH [74]. CXCR7 is considered a decoy receptor for CXCR4. CXCR7-deficient mice die one week after birth due to cardiovascular malfunction [75]. In this study, lacZ knock-in mice with the CXCR7 gene replaced by the lacZ gene displayed high LacZ expression in the vascular endothelial cells of the lung, indicating negative regulation of endothelial cell proliferation. The ligands of CXCR4 and CXCR7 are CXCL12 and CXCL11, respectively, but CXCR7 may associate with CXCL12 for negative regulation if CXCR7 works as a decoy receptor for CXCL12. Another study, however, described that treatment with a CXCR7 inhibitor alone did not attenuate PAH, while a combination of CXCR4 and CXCR7 inhibitors synergistically ameliorated PAH development [72].

A recent report revealed that hypoxia up-regulates interleukin (IL)-6 in the lung, and that IL-6 and IL-21 promote the differentiation of alveolar macrophages to M2 macrophages. Soluble factors such as CXCL12 can promote the proliferation of pulmonary artery smooth muscle cells, which can result in PAH [76]. We have demonstrated that, although there was no significant difference in the SDF-1 expression levels between the control group and PAH using a rat model, there were significant differences in the CXCR4 expression levels between these two groups, suggesting that CXCR4 signaling is also involved in PAH development [77]. Interestingly, smooth muscle cell (SMC)-specific phosphatase and tensin homolog (PTEN) deficient mice display an aorta structure similar to PAH, and SDF-1/CXCR4 is upregulated. Additionally, blocking by SDF-1 antibody can inhibit the growth of SMCs, suggesting PTEN/Akt enhanced growth through the SDF-1/CXCR4 pathway [78]. 

We investigated the effect of the CXCR4 inhibitor, silibinin, in the treatment of PAH and obtained evidence of treatment benefit. Silibinin is derived from the seeds of the milk thistle plant *Silybum marianum* L. [79]. Silibinin functions as an antioxidant and has been used to treat liver diseases [80]. Anticancer efficacy has also been reported [81,82,83,84,85,86,87,88,89,90,91,92,93,94]. Silibinin is associated with the down-regulation of nuclear factor-kappa B (NF-κB) and the inhibition of tumor necrosis factor-alpha (TNFα) through binding to IKKα [88]. Silibinin inhibits not only NF-κB but also other signal transduction pathways such as epidermal growth factor receptor and/or insulin growth factor receptor pathways to trigger apoptosis [83,85,86,95]. Silibinin can also inhibit the oxidative stress that causes cell injury or death and inflammatory reactions. Therefore, it can be used to protect against liver, heart, and lung damage. Silibinin mediates SDF-1-induced CXCR4 activation in tumor cells and exerts enhanced CXCR4 antagonistic effects when compared to AMD3100 [96]. Silibinin was reported to reduce inflammation associated with rheumatoid arthritis by reducing fibroblast-like apoptosis mediated by the suppression of NF-κB [97]. Silibinin is thought to function differently from AMD3100 and to influence different signal transduction pathways (Figure 1). More investigation is required to clarify these diverse effects. 

Silibinin has been traditionally used in folk medicine and is considered relatively safe. The toxic effect of long-term (2 years) usage of silymarin was previously investigated in rats and mice. Silymarin is an extract from the milk thistle plant. The main component is silibinin [79]. No significant differences were evident in longevity between the control (no silymarin extract exposure) and silymarin-exposed (12,500, 25,000, and 50,000 ppm silymarin) groups; however, the body weights were at least 10% lower in the 50,000 ppm silymarin group than in the control group after 9 weeks of treatment for males and 12 weeks of treatment for females [98]. Hepatocellular adenoma was significantly reduced in the treated rats [91,98].

### 3.3. WHIM Syndrome

WHIM syndrome is a rare inherited primary immunodeficiency disorder caused by autosomal dominant gain-of-function mutations in the *CXCR4* gene [99,100,101]. WHIM syndrome is also characterized as neutropenia associated with hyperplasia of mature neutrophils in the bone marrow, termed myelokathexis, and lymphopenia that reduces mature B and T cells. The *CXCR4* mutations associated with this syndrome occur in the C-terminal region of the receptor. These mutations prevent CXCR4 internalization in response to SDF-1 binding, which impairs SDF-1-induced receptor down-regulation, and increases CXCR4 signaling [99,101]. Since Plerixafor is a CXCR4 antagonist, it is expected to improve the prognosis of this disease. Very recently, it was reported that long-term, low-dose plerixafor treatment ameliorated WHIM syndrome and a hematological disorder in the bone marrow [102]. Because plerixafor is approved only for mobilizing hematopoietic stem cells currently, studies for optimizing the dosage amount and duration for WHIM treatment are required.

## 4. CXCR4 Involvement in Normal and Abnormal Development

### CXCR4 Signal Transduction and the CXCR4 Family

CXCR4 is a seven-transmembrane G protein-coupled receptor. Its promoter activity is upregulated by c-Myc and downregulated by Yin-Yang [103]. CXCR4 is involved in several physiological processes. Binding of SDF-1 to the extracellular domain of CXCR4 causes a conformational change that allows the chemokine to tightly bind the receptor pocket. Next, a second conformational change occurs to activate the intracellular trimeric G protein through the dissociation of the Gα subunit from the Gβ/Gγ dimer [104]. Once this activation occurs, cAMP stimulates the Src family of tyrosine kinases that activate the Ras, Raf, mitogen-activated protein kinase (MEK), and ERK pathways; however, many pathways exist, including the c-Jun [3], NF-κB [97], and mTORC2 [105] pathways. The direct role of CXCR4 in tissue repair or malignancy remains to be determined. Interestingly, this subgroup of CXC receptors control one another. As described previously, CXCR7 may be positively associated with CXCR4 and negatively associated with SDF-1. This change of partner may influence the CXCR subtype function; however, the CXCR4 and CXCR7 downstream environment may be complicated. Chen et al. reported that SDF-1-mediated cardiac stem cell migration can be inhibited by the application of small interfering RNA targeting CXCR4 or CXCR7, and this inhibition involves the decrease of phospho-ERK or phospho-Akt, respectively [106]. The Akt inhibitor MK2206 increases phospho-ERK and Raf-1. Given this, cross-talk may occur.

Recently, a number of reports have revealed that heterodimers of CXCR4 and CXCR3 or CXCR7 possess numerous functions, including roles in cancer cell invasion. Jin et al. used colorectal cancer cell lines to investigate the association between metastatic behavior and CXCR3 and CXCR4heterodimerization. They found that CXCR3 could form heterodimers, and CXCR3 could cause CXCR4 to locate to the cell surface. Knock-down of CXCR3 reduced surface CXCR4 in vitro. They also confirmed in vivo that in SW620 cells, CXCR3 and CXCR4 are expressed highly, and knock-down of CXCR3-A and CXCR4 reduced cell metastasis [107]. Song et al. reported that heterodimers of CXCR4 and CXCR7 enhanced tumorigenesis in transgenic mice. Additionally, they confirmed heterodimerization using co-immunoprecipitation [108]. These recent results suggest that heterodimerization of CXCR4 is important for signal transduction.

The role of the CXC chemokine family in lung cancer development has been previously investigated. A statistically significant decrease in the expression of CXCL4 and CXCL5 and a statistically significant increase in the expression of CXCL7 were observed; however, there was no significant difference in SDF-1 expression [109]. Conversely, the activation of CXCR4 in non-small-cell lung cancer [110] suggests that there is a cell-type specific enrollment of subtypes in different cancers.

The recent review by Messina et al. discussed the EMT/ mesenchymal-epithelial transition (MET) in terms of stemness, regeneration, and oncogenesis. The authors connected stemness and oncogenesis to the states of EMT and MET, and they also connected stem cells and CSCs [111]. SDF-1/CXCR4 is likely involved in this process. Indeed, deletion of the CXCR4 CTD results in EMT-like changes [58].

Taken together, SDF-1/CXCR4 signaling can exert counteracting effects and may be active in the (micro)environment surrounding stem cells, such as niches, which are commonly used by both normal stem cells and CSCs. In the case of stem/progenitor cells, they may migrate from the niches (bone marrow) to the SDF-1-expressing damaged tissues. The cells may be involved in tissue repair or dysplasia, which may promote cancer, as summarized in Figure 2. CSCs may be involved in metastasis. Additionally, SDF-1/CXCR4 may function in the upstream signal transduction of cytokines and EMT-like changes. The downstream signal transduction may be altered as a result of the amount or the nature of binding of SDF-1 to CXCR4.

## 5. Conclusions

CXCR4 is an interesting molecule that is associated with HIV infection, cancer, stem cell proliferation, differentiation, and migration. In the early stages of development, SDF-1 and CXCR4 function in cellular migration. Throughout life, particularly in adulthood, SDF-1/CXCR4 signaling may influence stem cell migration from the bone marrow or niche to repair damaged tissues. Increases in the number of CXCR4^+^ cells may be involved in inflammation/stem cell migration normally, and cancer/malignancy abnormally. Therefore, the amount and nature of binding of SDF-1 to CXCR4 may exert profound effects on downstream signal transduction, and these effects should be clarified in future studies.

## Figures and Tables

**Figure 1 cells-08-00185-f001:**
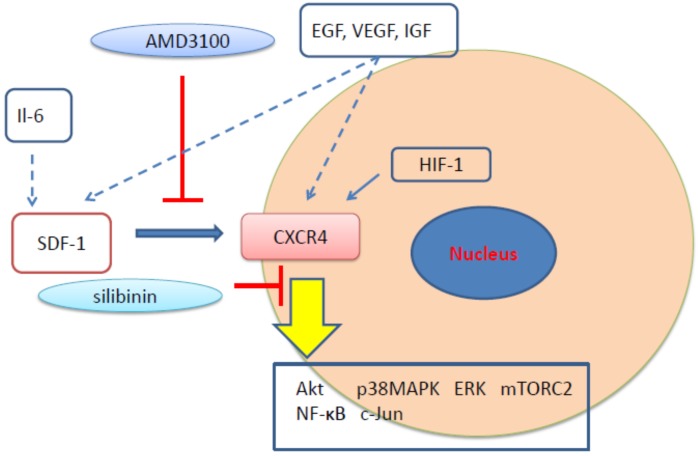
The sites of signal transduction for CXC motif chemokine receptor type 4 (CXCR4) antagonists.

**Figure 2 cells-08-00185-f002:**
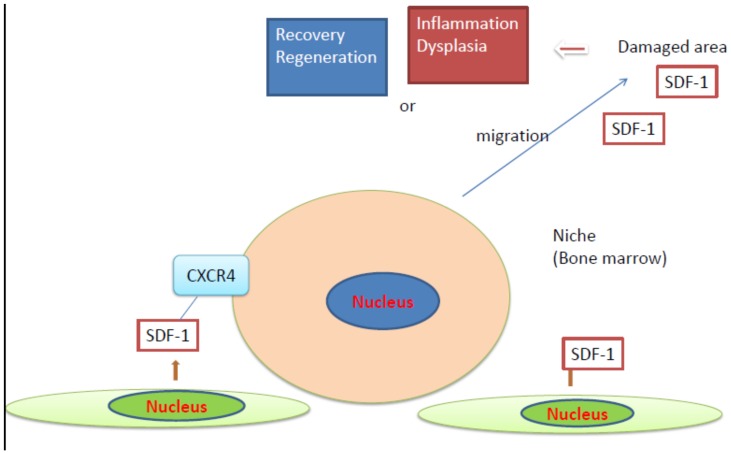
Action of stromal cell-derived factor (SDF)-1/CXCR4 axis in damaged tissues.

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
