# Peer review of "Involvement of CXCR4 in Normal and Abnormal Development"

_cells, 2019, doi:10.3390/cells8020185_

Round 1

Reviewer 1 Report

Involvement of CXCR4 in normal and/or abnormal  development

Nanako Kawaguchi, Ting-Ting Zhang, Toshio Nakanishi

General comments

The manuscript concerns on the role of CXCR4/SDF-1 signaling in development. The issue is interesting and scientifically important. However, the level of manuscript is still unsatisfactory.

The Abstract is still confusing and contains many mistakes. The “Introduction” chapter is confusing and chaotic. Moreover the Author use to much mental shortcuts. Many information included in the text are misleading and should be more precisely explained. Figures are not described. In current version, in my opinion, the manuscript still requires a lot of work to be published.

Detailed comments

1.      Line “11” – what does it mean “bone marrow trough niches”.

2.      Line “13” – “SDF/CXCR4 have been implicated in the maintenance of stemness maintenance and/or differentiation and so tissue-derived stem cells can be associated with SDF-1 and CXCR4 activity”. These sentence is completely confusing.

3.      Line “78-89” – quite incomprehensible.

4.      Line “111-114” – quite incomprehensible.

5.      Line “115-124” – chaotic.

6.      Line “127” “ In contrast to its natural role in stem cells during development, CXCR4 is also expressed in various cancers [47].” Why the Authors claim that cancer cells are not natural?

7.      Line “174”: “PAH is potentially fatal disease” – what does it mean? Moreover the abbreviation “PAH” has not been clarified.

8.      Line “103-108” – This fragment is completely incomprehensible.

9.      Line “234-268”- This fragment does not contain any information about dimers formed by CXCR4 with CXCR3 and CXCR7 that is very important for signal transduction and about other CXCR4 ligands.

Author Response

To reviewer 1

Thank you very much for the constructive comments. I really appreciate them, and I have tracked the revisions in the revised manuscript.

General comments

The manuscript concerns on the role of CXCR4/SDF-1 signaling in development. The issue is interesting and scientifically important. However, the level of manuscript is still unsatisfactory.

The Abstract is still confusing and contains many mistakes. The “Introduction” chapter is confusing and chaotic. Moreover the Author use to much mental shortcuts. Many information included in the text are misleading and should be more precisely explained. Figures are not described. In current version, in my opinion, the manuscript still requires a lot of work to be published.

I corrected the abstract, introduction, text, and figure legends.

Detailed comments

1.      Line “11” – what does it mean “bone marrow trough niches”.

I corrected this as “through niches in the bone marrow” (Lines 11-12, revised version).

2.      Line “13” – “SDF/CXCR4 have been implicated in the maintenance of stemness maintenance and/or differentiation and so tissue-derived stem cells can be associated with SDF-1 and CXCR4 activity”. These sentence is completely confusing.

I corrected this as “maintenance and/or differentiation of stemness” (Lines 13-14, revised version).

3.      Line “78-89” – quite incomprehensible.

I corrected this as follows (Lines 82-106, revised version):

DiGeorge or 22q11.2-deletion syndrome (DGS) is the most common human genetic deletion syndrome, and this disease results in mental disorders, craniofacial dismorphogenesis, thymus hypoplasia, and congenital heart defects caused by cardiac outflow anomalies. The responsible gene is located within 22q, which also includes T-box-containing transcription factor (TBX) 1 . Tbx-1 is one of the major genes responsible for forming the pharyngeal arch-derived structure [23-26]. Neural crest (NC) cells migrate from the dorsal hindbrain toward arches and ultimately form the face, neck, and chest. Additionally, NC cells contribute to cardiovascular structure. Therefore, DGS was believed to result from defective NC development; however, the TBX1 gene is not expressed in NC cells in pharyngeal arches. TBX1 is expressed in pharyngeal ectoderm, endoderm, and mesoderm. The interaction of pharyngeal NC cells and the surrounding tissues such as pharyngeal ectoderm, endoderm, and mesoderm were studied in the context of Tbx-1 signaling. Fibroblast growth factor 8, vascular endothelial growth factor, and more recently, SDF-1, were shown to be involved in this interaction [27-28]. Notably, the phenotype of an SDF-1/CXCR4 knock-out mouse is similar to that of a mouse exhibiting DGS, with defects in heart development and mental retardation [29]. Interestingly, SDF-1-expressing cells are present in the outflow track, while CXCR4+ cells are present in the NC. Thus, CXCR4 may be involved in the migration of NC cells during normal development. Recently, however, SDF-1/CXCR4 was suggested to function in cardiovascular development in the context of second heart field to endothelial cells and not in Tbx-1 haplosufficient arch artery phenotype [30]. More studies are required to clarify the CXCR4 contribution to NC cells in the context of normal development.

4.      Line “111-114” – quite incomprehensible.

I corrected this as follows (Lines 130-134, revised version):

Similarly, neural stem/progenitor cells are controlled by SDF-1/CXCR4 to maintain stemness [45]; however, CXCR4 activation promotes the differentiation of human embryonic stem cells into NSCs [46], suggesting that CXCR4 may work to stabilize NSCs. This form of up- or down-regulation to maintain certain populations such as NSCs may also occur in other stem cell populations.

5.      Line “123-138” – chaotic.

I corrected this as follows (Lines 135-151, revised version):

It is established that after pneumonectomy, the left lung possesses greater capacity for respiratory function due to alveolar regeneration. Recently, platelets were reported to be involved in this regeneration through SDF-1. Without platelets, which are induced by Thronbopoietin-deficient conditions (5% of wild type), regrowth of the lungs reduced markedly. SDF-1 could enhance this recovery significantly. Additionally, SDF-1+/+ platelets could rescue lung regeneration while SDF-1-/- platelets could not. Thus, pneumonectomized mice need SDF-1 from platelets for alveolar regeneration [47]. The recent idea that the lung features hematopoietic niches for hematopoietic stem cells is noteworthy [48]. Further studies focused on the involvement of SDF-1 and CXCR4 in regeneration could provide insight into their roles in this process. Notably, it has been found that the lung can possess niches for hematopoietic progenitors [49]. The liver is also thought to possess niches formed by hepatic stellate cells [50]. The niche is on the basement membrane surrounded by the matrix (laminin and collagen type IV). The stellate cells can act as myofibroblaststhat form a niche in bone marrow and can synthesize matrix. Interestingly, Jog 1, which is often expressed in the niche, is also expressed in these hepatic stellate cells. Of note, the stellate cells can be stem cells [50]. The results suggest a complicated flexible signature of the stem cell=containing niches.

6.      Line “127” “ In contrast to its natural role in stem cells during development, CXCR4 is also expressed in various cancers [47].” Why the Authors claim that cancer cells are not natural?

I changed natural to normal (Line 154)

7.      Line “174”: “PAH is potentially fatal disease” – what does it mean? Moreover the abbreviation “PAH” has not been clarified.

It was in the line 53. But I added the abbreviation of PAH as follows (Line 203-207, revised version)

Pulmonary arterial hypertension (PAH) is a rare disease characterized by high blood pressure in the arteries of the lungs, caused by an endothelial cell disorder and the abnormal proliferation of smooth muscle cells. These associate with inflammation and finally leads to occlusion within the pulmonary artery, which in turn triggers right ventricle hypertrophy.

8.      Line “103-108” – This fragment is completely incomprehensible.

I corrected this as follows (Lines 118-129, revised version):

Mesenchymal stem cells (MSCs) promote proliferation and differentiation of c-kit+ cardiac stem cells via SDF-1/CXCR4 signaling. If treated with the CXCR4 antagonist AMD3100, cardiac stem cells derived from a murine postnatal cardiac explant differentiate into cardiac myocytes [34]. Therefore, MSCs may control the self-renewal and/or proliferation of c-kit+ cardiac stem cells through SDF-1/CXCR4 signaling [34]. The c-kit+ cardiac stem cells are thought to rescue cardiac damage from injury such as ischemia [35-37]. Contradictory data, however, draw into question if the c-kit+ cells can act as cardiac stem cells [38,39]. When c-kit+ cells were utilized for cardiac repair, there were positive effects on cardiac differentiation and the enhancement of cardiac myocyte survival [40-43]. SDF-1/CXCR4 can act as an upstream regulator of these c-kit+ cells  and may contribute to the repair of tissues directly [44] or indirectly through the action of MSCs as previously described [34].

9.      Line “234-268”- This fragment does not contain any information about dimers formed by CXCR4 with CXCR3 and CXCR7 that is very important for signal transduction and about other CXCR4 ligands.

I added the following (Lines 286-297, revised version):

Recently, a number of reports have revealed that heterodimers of CXCR4 and CXCR3 or CXCR7 possess numerous functions, including roles in cancer cell invasion. Jin et al. used colorectal cancer cell lines to investigate  the association between metastatic behavior and CXCR3 and CXCR4 association. They found that CXCR3 could form heterodimers, and CXCR3 could cause CXCR4 to locate to the cell surface. Knock-down of CXCR3 reduced surface CXCR4 in vitro. They also confirmed in vivo that in SW620 cells, CXCR3 and CXCR4 are expressed highly, and knockdown of CXCR3-A and CXCR4 reduced cell metastasis [106]. Song et al. reported that heterodimers of CXCR4 and CXCR7 enhanced tumorgenesis in transgenic mice. Additionally, they confirmed heterodimerization using co-immunoprecipitation [107]. These recent results suggest that heterodimerization of CXCR4 is important for signal transduction.

Reviewer 2 Report

This revised manuscript is improved significantly after review. Overall, manuscript is well written and it covers comprehensively the role of CXCR4 in normal and disease state. 

Author Response

Thank you very much for the warm comments.

This revised manuscript is improved significantly after review. Overall, manuscript is well written and it covers comprehensively the role of CXCR4 in normal and disease state. 

Round 2

Reviewer 1 Report

-